# Exposure of *Mycobacterium abscessus* to Environmental Stress and Clinically Used Antibiotics Reveals Common Proteome Response among Pathogenic Mycobacteria

**DOI:** 10.3390/microorganisms8050698

**Published:** 2020-05-09

**Authors:** Rajoana Rojony, Lia Danelishvili, Anaamika Campeau, Jacob M. Wozniak, David J. Gonzalez, Luiz E. Bermudez

**Affiliations:** 1Department of Biomedical Sciences, Carlson College of Veterinary Medicine, Oregon State University, Corvallis, OR 97331, USA; rojonyrajoana@gmail.com; 2Department of Pharmacology, School of Medicine, Skaggs School of Pharmacy and Pharmaceutical Sciences, University of California San Diego, San Diego, CA 92093, USA; acampeau@ucsd.edu (A.C.); jakewozniak@gmail.com (J.M.W.); djgonzalez@ucsd.edu (D.J.G.); 3Department of Microbiology, College of Sciences, Oregon State University, Corvallis, OR 97331, USA

**Keywords:** *M. abscessus*, *M. avium*, proteomics, amikacin, linezolid, biofilm, anaerobic condition, antibiotic susceptibility, tolerance

## Abstract

*Mycobacterium abscessus* subsp. *abscessus* (MAB) is a clinically important nontuberculous mycobacterium (NTM) causing pulmonary infection in patients such as cystic fibrosis and bronchiectasis. MAB is naturally resistant to the majority of available antibiotics. In attempts to identify the fundamental response of MAB to aerobic, anaerobic, and biofilm conditions (as it is encountered in patients) and during exposure to antibiotics, we studied bacterial proteome using tandem mass tag mass spectrometry sequencing. Numerous de novo synthesized proteins belonging to diverse metabolic pathways were found in anaerobic and biofilm conditions, including glycolysis/gluconeogenesis, tricarboxylic acid (TCA) cycle, oxidative phosphorylation, nitrogen metabolism, and glyoxylate and dicarboxylate metabolism. Upon exposure to amikacin and linezolid under stress environments, MAB displayed metabolic enrichment for glycerophospholipid metabolism and oxidative phosphorylation. By comparing proteomes of two significant NTMs, MAB and *M. avium* subsp. *hominissuis*, we found highly synthesized shared enzymes of oxidative phosphorylation, TCA cycle, glycolysis/gluconeogenesis, glyoxylate/dicarboxylate, nitrogen metabolism, peptidoglycan biosynthesis, and glycerophospholipid/glycerolipid metabolism. The activation of peptidoglycan and fatty acid biosynthesis pathways indicates the attempt of bacteria to modify the cell wall, influencing the susceptibility to antibiotics. This study establishes global changes in the synthesis of enzymes promoting the metabolic shift and enhancing the pathogen resistance to antibiotics within different environments.

## 1. Introduction

The incidence and prevalence of patients with nontuberculous mycobacterial (NTM) infections has risen in recent years [1,2]. NTM strains of *Mycobacterium abscessus* complex (*M. abscessus* subsp. *abscessus* (MAB)*, M. abscessus* subsp. *bolletii and M. abscessus* subsp. *massiliense)* are rapidly growing opportunistic human pathogens ubiquitously found in the environment, and are capable of causing a wide range of clinical diseases in both immunocompetent and immunocompromised hosts [3]. Particularly, MAB is a leading pulmonary infection in patients with underlying structural lung diseases such as bronchiectasis, emphysema, and cystic fibrosis [3]. While MAB is a commonly isolated species from respiratory specimens, it can also cause wide spectrum skin, ocular, and central nervous system infections, bacteremia, and diseases in almost every human organ [1]. MAB is characterized by extremely high intrinsic resistance to anti-tuberculosis and most antimicrobial agents [3,4], and is responsible for significantly higher fatality rates than any other rapidly growing mycobacteria [5,6].

The list of antibiotics used for MAB infections in clinics is very short. The treatment of MAB infection relies on a combinational therapy with an aminoglycoside (amikacin) or a macrolide (clarithromycin) alongside cefoxitin, imipenem, and linezolid for several months [5], clearing infections mainly by targeting bacterial protein synthesis via binding to different rRNA subunits, or by disrupting the cell wall synthesis [7]. Linezolid is an antibiotic used for the treatment of drug-resistant infections caused by Gram-positive bacteria [8,9], and recent studies have suggested employing the drug as an alternative option for treatment of the multidrug resistant tuberculosis as well [10,11]. Despite multi-drug regimens designed for MAB patients, it is common in clinics to observe individuals who either do not respond to therapy, or respond only partially with a success rate of 25% to 42% [12]. The natural resistance to the majority of available antimicrobials and inability of drugs to rapidly kill MAB, even at bactericidal concentrations, are major clinical challenges for MAB treatment.

MAB, similar to other mycobacterial pathogens, can switch to a nonreplicating persistent state inside the host that can promote the development of antibiotic tolerance phenotype [13]. Experimental evidence suggests that low concentration of nutrients, low pH and lack of oxygen stimulates the development of bacterial nonreplicative state and influences the selection of MAB subpopulation with an intrinsic resistance mechanism [14]. Bacterial drug-tolerance phenotype is a result of altered metabolic state and changes on the cell surface of MAB, which subsequently affects the cell wall permeability and impairs drug penetration [12]. MAB can survive in low oxygen conditions within lung granulomas of cystic fibrosis patients by shifting the aerobic metabolic state to anaerobic, and by inducing a biofilm phenotype [14,15,16]. A high tendency of MAB to form biofilms in patients’ lung airways is an equally important factor that influences the pathogen killing by antibiotics. The reduced drug permeation, often due to biofilm structures, significantly minimizes the effect of antibiotic action [15,17].

The discovery of new treatment strategies is needed to fight against significant NTMs, including MAB. In order to fill an important gap in fundamental knowledge on how MAB survive under nonreplicating and biofilm conditions, and how this phenotypic remodeling promotes antibiotic tolerance and increases bacterial survival, we investigated MAB proteome response under aerobic, anaerobic, and biofilm conditions, and compared these changes to bacterial responses under exposure to bactericidal concentrations of active antimicrobials under the same conditions. 

In this study, we also compared the proteomic profiles of two NTM species, MAB and *M. avium* subsp. *hominissuis* (MAH), to identify shared pathways possibly regulating changes that mycobacteria undergo within similar conditions and during antibiotic treatment. 

## 2. Materials and Methods

### 2.1. Mycobacteria Strains and Culture Conditions

A clinical isolate of *Mycobacterium abscessus subsp. abscessus* 19,977 with a smooth colony phenotype was purchased from the American Type Culture Collection (ATCC; Gaithersburg, MD, USA) and maintained either in 7H9 Middlebrook broth or on 7H10 Middlebrook agar supplemented with 10% oleic acid, albumin, dextrose, and catalase (OADC; Hardy Diagnostics, Santa Maria, CA, USA) at 37 °C till mid-log phase (2–3 days). Bacterial inoculum was prepared in Hanks’ balanced salt solution (HBSS; VWR, Visalia, CA, USA), and visually adjusted to a McFarland 0.5 standard equivalent to 1.5 × 10^8^ colony forming unit (CFU)/mL cell density. For precise calculations, the inoculum was serially diluted, plated on 7H10 agar plates and CFU counts were recorded after 7 days of incubation. *Mycobacterium smegmatis* strain mc^2^ 155 were used to construct overexpression clones of MAB and MAH genes, and kanamycin (Sigma-Aldrich, Saint Louis, MO, USA) at 50 μg/mL concentration was added to *M. smegmatis* cultures, where it was appropriate.

### 2.2. Antimicrobial Compounds and Susceptibility Testing

Amikacin (AMK) was purchased from Sigma-Aldrich (Saint Louis, MO, USA) and linezolid (LNZ) from Biomol GmbH (Hamburg, Germany). To create stock solutions, AMK were solubilized in water and linezolid in DMSO. The working drug concentrations were obtained by further dilution of the antibiotic stock solutions in the HBSS. 

The antibiotic susceptibility test was performed using a broth microdilution method in the concentration range of 0.065 to 256 µg/mL for both antibiotics. Briefly, 3 × 10^6^ CFU/mL were cultured in 7H9 Middlebrook broth with and without antibiotics and incubated in a shaker at 37 °C. The minimal inhibitory concentration (MIC) and bactericidal concentration (BC) were visually determined at day 6. In addition, BC tubes were centrifuged at 10,000× *g*, resuspended in HBSS and plated on 7H10 agar plates to check bacterial viability, if any. The drug concentration that inhibited 99.9% bacterial growth was considered as BC.

### 2.3. Antibiotic Killing Kinetics in Vitro

MAB inoculum of 3 × 10^6^ bacteria/mL was cultured in 7H9 liquid media supplemented with bactericidal concentrations of AMK (32 μg/mL) and LNZ (128 μg/mL). Samples were tested for bacterial CFU counts 1, 2, 4, and 6 days post-infection by plating the serial dilutions on 7H10 agar plates. 7H9 broth without antibiotics served as the bacterial growth control. 

To determine antibiotic killing kinetics against MAB in anaerobic conditions, 3 × 10^6^ bacteria/mL were inoculated into 3mL of 7H9 broth with and without antibiotics. Bacterial cultures were placed into anaerobic jars (BD BBL™ GasPak™ Jar; Franklin Lakes, NJ, USA) together with the anaerobic CO_2_ indicators also obtained from BD Company. After sealing the jar with the grease, one jar for each time point was placed in the shaker incubator with agitation at 30 rpm and at 37 °C. Tubes were removed at each time point, centrifuged at 3500 rpm for 30 min, and then MAB pellets were resuspended in HBSS. Serial dilutions were cultured on 7H10 agar plate for CFU counts. The killing kinetic under the biofilm conditions were determined as follows: 100 µl of 3 × 10^7^ bacterial inoculum per milliliter was distributed into 96-well plates and maintained for 7 days at 25 °C. A week later, supernatants were removed and replenished with fresh HBSS either containing bactericidal concentration of AMK and LNZ, or no antibiotic. MAB biofilms were disrupted at selected time points, and serial dilutions were plated for viable bacterial counts. Experiments were performed in three biological replicates and carried out in duplicate for each condition, including controls.

### 2.4. Antibiotics Killing Kinetics in Human Macrophage

Human THP-1 cell line (ATCC, TIB-202) was maintained in RPMI 1640 medium supplemented with heat-inactivated 10% fetal bovine serum (FBS; Gemini Bio, Sacramento, CA), L-glutamine, and 25 mM HEPES (Corning; Corning, NY, USA) at 37 °C and in an atmosphere of 5% CO_2_. Intracellular killing assays were performed as previously described, with minor modifications [18]. Briefly, THP-1 monocytes were treated with 20 ng/mL of Phorbol 12-myristate 13-acetate (PMA; Sigma-Aldrich, Saint Louis, MO, USA) to stimulate differentiation into macrophages. Approximately, 3 × 10^5^ cells per well were added to a 48-well tissue culture plate with PMA for 24 h, and then rested for an additional 48 h in fresh RPMI medium without PMA. 

Monolayers were infected with MAB at a multiplicity of infection (MOI) of 5 bacteria to 1 cell. After 2 h, cells were washed three times with HBSS followed with 1h AMK treatment (100 μg/mL) to remove extracellular bacteria. Infected THP-1 monolayers were treated with AMK 32 μg/mL, LNZ 128 μg/mL or no antibiotic, and replenished with new media (with and without antibiotics) every other day. THP-1 cells were lysed at 2 h to record invasion rates. To determine number of survival/viable intracellular bacteria, lysates were obtained at day 1, 3, 5, and 7, and serial dilutions were plated on 7H10 agar plates. To prepare inoculums of anaerobic or biofilm condition, MAB was either pre-incubated in the anaerobic chamber for 24 h at 25 °C or cultured for 7 days to form biofilms as described above. THP-1 cells were infected with MOI of 5 bacteria to 1 cell for 2 h and processed to remove extracellular bacteria. Monolayers were replenished with fresh media every other day with or without antibiotics, and lysed at 2 h (baseline), 1, 3, 5, and 7 days post-infection for viable bacterial CFU counts.

### 2.5. Protein Sample Preparation for Sequencing

MAB (approximately 5 × 10^8^ CFU/mL) was inoculated in 50 mL of 7H9 broth with or without bactericidal concentrations of AMK or LNZ under aerobic and anaerobic conditions at 37 °C. While in the aerobic setting tubes were placed into a shaker under agitation at 200 rpm, for anaerobic assay samples were placed in the anaerobic jar with gentle agitation at 30rpm. Bacteria were lysed after 24 h exposure with antibiotic. For the biofilm study, MAB inoculum of 9 × 10^8^ CFU/mL in HBSS were adjusted to the McFarland standard #3 and to the absorbance ranging 0.38–0.42 at 625 nm. MAB was cultured in the 75 cm^2^ tissue culture flasks at 25 °C for 7 days at which point the supernatants were gently removed without disturbing the biofilm structures. Flasks were replenished with new medium containing AMK, LNZ, or no antibiotics. After 24 h, samples were centrifuged at 3600× *g* for 20 min at 4 °C, bacterial pellets were washed with HBSS and lysed with 3% Sodium Dodecyl Sulfate (SDS; Sigma-Aldrich, Saint Louis, MO, USA) containing EDTA-free protease inhibitor cocktail (Sigma-Aldrich, Saint Louis, MO, USA). In addition, samples were mechanically disrupted through bead-beating, and then cleared by centrifugation at 15,000× *g* for 10 min and through filtration with 0.22 µm filters. Total protein concentrations were determined using the Thermo Scientific NanoDrop device.

### 2.6. Tandem Mass Tag (TMT) Labeling and Liquid Chromatography–Mass Spectrometry (LC–MS)

Quantitative mass spectrometry analysis of sample lysates was performed using previously described methods [19,20,21]. Briefly, samples were subjected to probe sonication-assisted lysis in 4M urea with 50 mM HEPES, pH = 8.5 and 3% SDS lysis buffer. Disulfide bond reduction was performed in 5mM dithiothreitol (DTT) at 56 °C for 30 min. Free cysteine residues were alkylated in 15 mM iodoacetamide in the dark for 15 min, and the reaction was quenched for 20 min in 5mM DTT for 20 min. Protein samples were precipitated using a trichloroacetic acid (TCA; Sigma-Aldrich, Saint Louis, MO, USA) precipitation protocol. TCA was added to samples on ice and protein was pelleted via centrifugation. Samples were washed twice with ice-cold acetone. Samples were dried and resuspended, and proteins were digested in LysC overnight and in trypsin at 37 °C for 6 h. Samples were acidified and desalted on C18 columns using previously-described methods [22]. 

Peptide quantification was performed using the Pierce Quantitative Colorimetric Peptide Assay (ThermoFisher Scientific, Waltham, MA, USA). Fifty µg of each sample and pooled internal standard was labeled with tandem mass tags (TMTs) [23,24]. Following labeling, 10-plex samples were mixed and desalted on C18 columns. 10-plexes were fractionated using reverse-phase liquid chromatography. Alternating 96 resultant fractions were lyophilized [25]. MS analysis was performed on an Orbitrap Fusion Tribrid Mass Spectrometer with in-line Easy nLC System at University of California San Diego. 

### 2.7. MS Data Processing

Raw data files were searched using Proteome Discoverer 2.1 with SEQUEST-HT using previously-described methods with modifications as appropriate [26,27,28,29]. Briefly, files were searched against the MAB 19,977 strain reference proteome. The precursor and fragment ion mass tolerances were 50 ppm and 0.6 Da, respectively. Methionine oxidation (+15.995 Da) was used as a dynamic modification and isobaric tandem mass tags at the N-termini and on lysine residues (+229.163 Da) and carbamidomethylation of cysteine residues (+57.021 Da) were used as static modifications. A 1% false discovery rate (FDR) threshold was used.

High confidence peptide spectral matches (PSM) were filtered to exclude spectra with “rejected” PSM ambiguity, isolation interference greater than 25 and average relative abundance less than 10. TMT relative abundance values were summed to the protein levels. Data were normalized to the internal standard divided by the median of all internal standards and next against median signal for each label divided by the median of all median values [30]. The proteomics data used in this manuscript were deposited to ProteomeXchange through MassIVE. The MAH data can be found under the identifier PXD018956 and for MAB under the identifier PXD018957.

We categorized MAB proteins into functional groups of *Mycobacterium tuberculosis* strain H37Rv because they are well defined for this pathogen but not for MAB. The amino acid sequences of MAB proteins enriched with ≥1.5-fold were blasted using the *M. tuberculosis* protein sequence database at the Institute Pasteur’s web server (http://genolist.pasteur.fr/TubercuList/). Proteins that did not display similarity were classified based on their predicted function or grouped into the hypothetical class of unknown proteins. 

### 2.8. Construction of MAB and MAH Gene Overexpression Clones in M. smegmatis

MAB and MAH homologous genes encoding enzymes of peptidoglycan, pantothenate, and CoA biosynthesis and nitrogen metabolism pathways that were synthesized more than ≥1.5-fol in both pathogens were selected. The single genes or operons of these pathways 1) the peptidoglycan biosynthesis pathway *MAB_2003* (Gene ID: 5964517) and *MAV_2333* (Gene ID: 4525877), 2) the pantothenate and CoA biosynthesis pathway *MAB_0541* (Gene ID: 5963078), *MAB_0542* (Gene ID: 5963079), *MAB_0543* (Gene ID: 5963080) and *MAV_0551* (Gene ID: 4528629), *MAV_0552* (Gene ID: 4530264), *MAV_0553* (Gene ID: 4529546) and 3) the nitrogen metabolism pathway *MAB_3521c* (Gene ID: 5966021), *MAB_3522c* (Gene ID: 5966022) and *MAV_4903* (Gene ID: 4530094), *MAV_4904* (Gene ID: 4530193) were amplified from MAB and MAH genomic DNAs and ligated into the pMV261 mycobacterial shuttle vector. The recombinant plasmids were sequenced, transformed into non-pathogenic *M. smegmatis,* and resulting bacterial clones were selected on 7H10 agar plates containing 50 μg/mL kanamycin [31].

*M. smegmatis* clones were tested for sensitivity to antibiotics in vitro and cultured macrophages. Briefly, for in vitro study, bacterial inoculum of 10^6^ CFU/mL was cultured in liquid medium containing minimal inhibitory concentration of the following drugs: AMK 1 μg/mL, clarithromycin (CLA) 2 μg/mL, or LNZ 1 μg/mL. Samples were taken after 1, 2, 4, 6 and 8 days of drug exposure, and serial dilutions were plated onto 7H10 agar plates for CFU determination. In the macrophage survival assay, differentiated THP-1 cell monolayers were infected with *M. smegmatis* control and experimental clones at MOI of 5 bacteria to 1 cell for 2 h. Monolayers were washed three times with HBSS and treated with 100 μg/mL AMK for additional 1h. Next, THP-1 macrophages were treated with either AMK 1 μg/mL, CLA 2 μg/mL, or LNZ 1 μg/mL or no antibiotic as a control. THP-1 cells were replenished with new media containing antibiotics, or without antibiotics every other day. Cells were lysed at 2 h (baseline), day 1, 2, 4, 6, and 8, followed by plating on 7H10 agar plates to determine the number of viable intracellular bacteria. All experiments were carried out in duplicate and repeated three times.

### 2.9. Statistical Analysis

Binary comparisons were performed in order to generate volcano plots in GraphPad Prism 7. Statistical significance was determined using the Student’s t-test with Welch’s correction as appropriate. Morpheus analysis was used for K-means clustering, and the elbow method for cluster optimization (https://software.broadinstitute.org/morpheus). 

The in vitro and macrophage assays were repeated at least three times. Comparisons among experimental and control groups were analyzed with ANOVA and Student’s t-test. Results are expressed as a mean ± standard deviation. The *p* value of < 0.05 was considered to be statistically significant.

## 3. Results

### 3.1. The Delayed Killing of MAB by Antimicrobials

AMK and LNZ are among the most common antibiotics used to treat MAB infections in clinics. The minimal inhibitory concentration (MIC) at which 90% of MAB growth was inhibited by the antibiotic was determined using the broth microdilution method, and was found 2 μg/mL and 8 μg/mL for AMK and LNZ, respectively. Antibiotic concentrations up to 16x the MIC of AMK and LNZ did not result in significant reduction of bacterial CFU at day 6. The bactericidal concentration (BC) at which 99.9% of MAB growth was inhibited by the antibiotic was recorded with the CFU assay in the liquid culture and was found 32 μg/mL and 128 μg/mL for AMK and LNZ, respectively. The antibiotic killing dynamics were investigated in aerobic, anaerobic, and biofilm conditions for up to 6 days (Figure 1A). A complete bacterial clearance was observed under aerobic conditions at day 4 following AMK or LNZ treatment. However, killing dynamics of anaerobic and biofilm conditions significantly differed from aerobic conditions. While a slight decline in viable bacteria were seen in the AMK treatment group under anaerobic conditions, LNZ treatment eliminated 100% of MAB at day 4. No significant changes were recorded in bacterial killing when the untreated control group of MAB biofilm was compared with either AMK or LNZ treatment for over 6 days (Figure 1A).

### 3.2. MAB Survival in Human Macrophages Following Exposure to Antimicrobials

THP-1 macrophage monolayers were infected with MAB expressing either aerobic, anaerobic, or biofilm phenotypes, and subjected to antibiotic treatment for 7 days. For each phenotype, infected cells that were not exposed to antimicrobials served as a control. As shown in Figure 1B, THP-1 cells had increased uptake of MAB of the biofilm phenotype when compared with invasion rates of aerobic and anaerobic bacteria. In the survival assay of aerobic condition, MAB was able to grow intracellularly within macrophages with no antibiotic treatment; whereas during exposure to bactericidal concentrations of AMK and LNZ, the THP-1 cells were unable to clear the infection (Figure 1C), resulting in 2.5- and 3.5-log decrease of intracellular bacteria at day 7 post-infection when compared with untreated cells, respectively. MAB in anaerobic conditions grew within macrophages similarly as the aerobic bacteria with and without exposure to antibiotics (Figure 1C). We observed that the biofilm phenotype enhanced MAB growth in cultured macrophages relative to other MAB phenotypes and, to highlight, the results represent combined CFUs of intracellular and extracellular bacteria (leaving infected cells) at given time points. AMK and LNZ treatments did not clear MAB of biofilm phenotype but exhibited killing effect and significantly delayed MAB growth (Figure 1C).

### 3.3. MAB Proteomics under Aerobic, Anaerobic and Biofilm Conditions with and without Exposure to Antimicrobials

Quantitative TMT mass spectrometric sequencing was used to analyze bacterial proteomics for experimental and control groups at the 24 h time-point. MAB proteomics data was deposited to ProteomeXchange through MassIVE and can be found under the identifier PXD018957. The supplemental data displays a list of overall proteins (approximately 4000) identified across all groups with their normalized abundance, annotations and fold changes over controls at corresponding time points for each environmental condition and antibiotic exposure. Volcano plots presented in Figure 2 give a global overview of induced and repressed bacterial proteins following exposure to different environmental conditions and antibiotics. More specifically, while the incubation in anaerobic and biofilm conditions resulted in enrichment of 551 and 866 proteins, the synthesis of 556 and 743 proteins were downregulated when compared to MAB of aerobic condition. Proteome analysis of MAB treated with AMK for 24 h revealed 587, 43, 139 synthesized and 437, 8, 254 downregulated proteins in aerobic, anaerobic, and biofilm conditions, respectively, when compared to control no drug treatment group. In the presence of LNZ, 755, 483, 72 proteins were enriched and 720, 53, 320 proteins were downregulated in aerobic, anaerobic and biofilm conditions, respectively, when compared with only condition controls. Figure 3 demonstrates the distribution of the average fold changes for proteins at different environmental conditions with or without drug treatments.

For a global overview of MAB proteome, we used Morpheus to identify K-means clustering (Figure 4A). Protein levels in AMK and LNZ treatments were paired and analyzed by antibiotic and no drug treatment under corresponding conditions. Furthermore, we found seven optimal clusters using the elbow method. While proteins of the clusters 1 mainly were expressed under aerobic conditions, the cluster 3 proteins were increased under biofilm conditions. The synthesis of cluster 2 and cluster 6 proteins was significantly enriched in aerobic conditions during exposure to AMK and LNZ, respectively. Pie charts of the Figure 4B demonstrate enrichment of metabolic pathways that are related to each cluster.

### 3.4. Functional Grouping of MAB Enriched Proteins

24 h exposure of MAB to AMK and LNZ under aerobic conditions triggered synthesis of 587 and 755 proteins, respectively, that was ≥1.5-fold higher than in control groups with no antibiotic exposure. In order to categorize differentially synthesized proteins into functional groups, amino acid sequences of identified bacterial proteins were blasted against well-studied *M. tuberculosis* H37Rv strain on the TubercuList web server. The hypothetical proteins that had no homology to *M. tuberculosis* were analyzed for domain/motif presence using the CD-BLAST network service (National Center for Biotechnology Information) and the Pfam database (Sanger Institute). This group of proteins was classified either by a predicted function or pooled into the hypothetical class with no known function.

In anaerobic and biofilm conditions, we found total of 551 and 866 proteins enriched with 1.5 and more fold over aerobic condition. The functional categories, respectively, are listed in parenthesis and are associated with the intermediate metabolism and respiration (227 and 362 proteins), cell wall and cell processes (101 and 155 proteins), regulatory proteins (43 and 92 proteins), lipid metabolism (36 and 40 proteins), virulence, detoxification and adaptation (51 and 49 proteins), metabolic enzymes belonging to oxidoreductase activity category (31 and 49 proteins), information pathway (26 and 23 proteins), and proteins with unknown function (84 and 155 proteins) (Figure 5A).

Most represented categories under aerobic conditions during exposure to both AMK and LNZ antibiotics were intermediate metabolism and respiration (477 proteins), cell wall and cell processes (298 proteins), regulatory proteins (118 proteins), lipid metabolism (107 proteins), virulence, detoxification, adaptation (96 proteins), metabolic enzymes falling into oxidoreductase activity category (72 proteins), information pathway (72 proteins) and majority of proteins with unknown function (234 proteins) (Figure 5B).

Under anaerobic conditions, AMK and LNZ induced the synthesis of 43 and 483 MAB proteins by ≥1.5-fold over control, respectively. The functional distribution of anaerobic categories during AMK and LNZ treatment, respectively, are as follows: 4 and 101 proteins of the intermediate metabolism and respiration; 21 and 79 of regulatory proteins; 5 and 15 proteins from the virulence, detoxification, adaptation group; 1 and 19 were enzymes with the oxidoreductase activity; 9 and 88 proteins in cell wall and cell processes; and 1 and 19 proteins from the information pathway (Figure 5C).

Represented categories for AMK and LNZ exposed bacteria under biofilm conditions were intermediary metabolism and respiration (26 and 27 proteins), regulatory proteins (28 and 4 proteins), virulence, detoxification and adaptation (4 and 1 proteins), cell wall and cell processes (33 and 17 proteins), lipid metabolism (8 and 2 proteins), information pathways (11 and 10 proteins), enzymes with oxidoreductase activity (6 and 1 proteins), and uncharacterized proteins (51 and 26 proteins), respectively (Figure 5D).

### 3.5. MAB Metabolic Pathways Expressed under Environmental Conditions and during Presence of Antibiotics

The protein assignment to KEGG (Kyoto Encyclopedia of Genes and Genomes) metabolic pathways is presented in the Figure 6. Eleven metabolic pathways were expressed under anaerobic and thirteen under biofilm condition when compared with the aerobic control alone (Figure 6A). Among them, ten pathways were common between anaerobic and biofilm conditions. The metabolic pathways present in both anaerobic and biofilm conditions are pyruvate metabolism, glycolysis/gluconeogenesis, citrate cycle (TCA cycle), oxidative phosphorylation, carbon metabolism, starch and sucrose metabolism, alanine, aspartate and glutamate metabolism, nitrogen metabolism and glyoxylate and dicarboxylate metabolism.

Ten metabolic pathways were highly expressed under aerobic condition, eight under anaerobic and six under biofilm condition when treated with AMK, and the expression of these pathways in anaerobic and biofilm conditions were significantly greater than the one seen in the aerobic conditions alone (Figure 6B). The oxidative phosphorylation pathway was more prominent in aerobic, anaerobic and biofilm conditions in the presence of AMK. The biofilm phenotype increases glycerophospholipid metabolism and fatty acid biosynthesis in MAB after exposure with AMK. Pyruvate metabolism, glycolysis/gluconeogenesis, arginine, proline, alanine, aspartate, and glutamate metabolism pathways were increased in aerobic condition due to AMK treatment. We also identified nine metabolic pathways highly expressed under aerobic condition, seven under anaerobic and four under biofilm condition when treated with LNZ (Figure 6C). In the presence of LNZ, MAB of aerobic, anaerobic and biofilm conditions were enriched with the oxidative phosphorylation pathway. While the glycerophospholipid metabolism was increased in MAB biofilms during exposure to LNZ, ABC transporters, butanoate metabolism, and proteasome metabolism were significantly increased in anaerobic conditions with LNZ treatment. In addition, an increase in glycolysis/gluconeogenesis, pyruvate metabolism, arginine and proline metabolism, non-homologous end-joining and atrazine degradation were observed in aerobic conditions due to LNZ.

### 3.6. Common Metabolic Pathways of MAB and MAH Expressed under Different Environmental Conditions and During Exposure with Antibiotics

MAB and MAH are two most common opportunistic mycobacterial pathogens causing pulmonary infections in patients with underlying structural lung diseases. In the current study, we aimed to compare proteomic profiles of two NTM species, and identify common pathways possibly regulating changes that bacteria undergo within similar environmental conditions and during antibiotic treatment. Our previous study established global changes in MAH proteomics upon exposure to conditions that the pathogen can encounter in the lung environment and during therapy with anti-MAH antibiotics [19]. With a goal to establish common pathways and identify targets in both NTM pathogens promoting the general mechanism of tolerance, we analyzed proteome response for each pathogen to environmental conditions and to antimicrobial agents.

It was observed that while incubation of MAH in anaerobic condition triggered synthesis of 409 proteins by ≥1.5-fold higher than aerobic condition [19], MAB proteome of anaerobic phenotype was enriched with 551 proteins in the same condition. The biological pathway mapping of MAH and MAB of anaerobic condition has identified nineteen metabolic pathways commonly expressed within the same environment (Figure 7A). We have discovered 603 proteins of MAH and 866 of MAB synthesized over 1.5-fold in biofilm conditions when compared to bacteria of aerobic phenotype, and these proteins were found to belong to nine common metabolic pathways (Figure 7B and [19]). Five metabolic pathways of oxidative phosphorylation, nitrogen metabolism, biosynthesis of secondary metabolites, peptidoglycan biosynthesis and glyoxylate dicarboxylate metabolism were shared between anaerobic and biofilm groups.

In aerobic condition, we have identified enrichment with a total of 642 MAH proteins when exposed to AMK and clarithromycin (CLA). Analysis of the MAB proteome within the same environmental conditions and during treatment with AMK and LNZ found that a total of 1342 proteins that were produced at ≥1.5-fold greater than the stress condition alone with no antibiotic treatment. The pathway analysis established nine shared metabolic pathways in both species, presented in Figure 7C. In order to modify the hydrophobic properties of the permeability barrier and change the cell wall structure, it was evident that in all tested conditions (anaerobic, biofilm, antibiotic treatments) both MAH and MAB upregulated the peptidoglycan biosynthesis pathway (Figure 8A). In addition, in anaerobic and biofilm experimental groups of both pathogens, numerous enzymes related to pantothenate and CoA biosynthesis pathway (Figure 8B) and nitrogen metabolism (Figure 8C) were highly synthesized. MAB and MAH homologous proteins are shown in Figure 8.

### 3.7. Antibiotic Killing Dynamics against M. smegmatis Clones Expressing MAB and MAH Genes in Vitro and in Macrophages

We hypothesized that homologues proteins of MAB and MAH that are highly synthesized in both anaerobic and biofilm environments and during exposure with antibiotics, and belong to metabolic pathways shared between both NTM pathogens most likely will to be associated with the persistence (tolerance) mechanism and will aid bacterial survival during antibiotic treatment. The bactericidal activity of anti-NTM drugs AMK, CLA, and LNZ were evaluated for over ten days using *M. smegmatis* protein overexpression clones. *In vitro* studies assessed in the 7H9 liquid culture medium, revealed no difference in *M. smegmatis* growth for the control clone containing a skeleton plasmid and clones expressing MAB and MAH genes (Figure 9). While AMK, CLA and LNZ had significant bactericidal activity against the control *M. smegmatis* at 2, 4, and 6 days, overexpression of *MAB_2003* and *MAV_2333* genes of the peptidoglycan biosynthesis pathway, *MAB_0541-MAB_0543* and *MAH_0551-MAV_0553* of the pantothenate and CoA biosynthesis pathway, and *MAB_3521c-MAB_3522c* and *MAV_4903-MAV_4904* of the nitrogen metabolism pathway had diverse outcomes and mostly delayed bacterial killing time either by two or four days (Figure 9). *MAB_2003* and *MAV_2333* are homologous genes (*mraY*) encoding for phosphor-N- acetylmuramoyl-pentapeptide-transferase. This enzyme catalyzes the first step of the lipid cycle reactions in the biosynthesis of the cell wall peptidoglycan. While the panC pantothenate synthetase (*MAB_0541* and *MAH_0551*) catalyze the condensation of pantoate with beta-alanine in an ATP-dependent reaction via a pantoyl-adenylate intermediate, the panD aspartate 1-decarboxylase (*MAB_0542* and *MAH_0552*) catalyzes the pyruvoyl-dependent decarboxylation of aspartate to produce beta-alanine. *MAB_0543* and *MAH_0553* (*coax*) belong to type III pantothenate kinases, initiating catalysis of the first step in CoA biosynthesis. *MAB_3521c/MAV_4903* genes and *MAB_3522c/MAV_4904* genes are homologs and encode for *nirD* and *nirB* (small and large) subunits of the nitrite reductase.

Likewise, we examined the drug activity against intracellular infection when THP-1 cells were infected with *M. smegmatis* clones. At first, we performed macrophage survival studies without any antibiotic treatment and identified that *M. smegmatis* control and the overexpressed clones had similar intracellular growth kinetics (Figure 10). In macrophage drug treatment studies, while infections by the *M. smegmatis* control and *MAV_4903-MAV_4904* clone of the nitrogen metabolism were cleared by all three antibiotics at day two post-infection, expression of the rest of MAB and MAH genes of the peptidoglycan biosynthesis, pantothenate, and CoA biosynthesis and nitrogen metabolism pathway had different effect on intracellular bacterial survival, and prolonged *M. smegamatis* growth for additional two to four days depending on the clone (Figure 10).

## 4. Discussion

Pulmonary diseases caused by *M. abscessus* subsp. *abscessus* (MAB) are extremely challenging to treat. The therapy involves combinational regimens with a minimum of three antibiotics and 18–24 months of duration to achieve some certain positive outcomes [3,32,33]. The fact is that the official regimens used for MAB treatment in clinics have a little or no evidence of in vivo efficacy [4] and, despite of combinational regimens, the success rate ranges from only 25% to 42% [12]. The natural ability of this pathogen to resist to the majority of available antibiotics that are currently used to treat other NTM pathogens and *M. tuberculosis* makes therapy efforts even more complex. For example, it has been shown that approximately 20% of MAB isolates of pulmonary infections fail to respond to macrolide therapy due to presence of the erythromycin ribosome methyltransferase (*erm*) gene [34,35].

An additional layer of complexity to clear MAB infection is the stimulation of the drug-tolerant/persistent phenotype by the host environmental stresses. The diverse conditions within the host facilitate metabolic and cell surface remodeling in bacterial [36], and these phenotypic changes subsequently influence the drug penetration process through the cell wall, decreasing antibiotic efficacy. MAB disease progression shares some aspects with *M. tuberculosis* and, as the disease evolves, this pathogen also lives inside granulomas or pulmonary nodules that are characterized with the low oxygen tension (anaerobic conditions) [37]. Furthermore, MAB has an ability to form biofilm on the surface of the airways inside of the human lung [38]. Due to the fact that the drug activity is generally established against aerobically growing and metabolically active bacteria, when the same antimicrobials are used against bacteria of different phenotypes, treatment outcomes vary, and many times they are ineffective. In fact, within granulomas and in biofilms of lung airways, majority of MAB remains in a non-replicating and low metabolic state. This is an entirely different environment than that in which antibiotic activities are tested. In order to define the relationship between MAB survival in either biofilm or anaerobic environments (as it is encountered in patient’s lung) and drug tolerance, this study aimed to characterize global proteome response of this pathogen within the biologically relevant environments and during exposure to antibiotics. Our results define numerous enzymes highly synthesized in MAB under different stress conditions, and associated with metabolic pathways regulated by this pathogen. Some changes have already been implied to aid bacterial survival and have been shown to radically modify the cell wall by obscuring surface targets. For example, surface localized secreted LprB lipoprotein that has been shown in MAH to decrease the pathogen uptake by macrophages and increase tolerance to aminoglycoside [19].

Our results demonstrate that MAB of biofilm and anaerobic phenotypes induces a central metabolic shift that contributes to the pathogen survival by upregulating ten common metabolic pathways under both stress environments. The increased level of the citrate cycle (TCA cycle), oxidative phosphorylation and glyoxylate dicarboxylate metabolism was observed under both conditions. The TCA cycle and increased glycoxylate shunt has been shown to provide the glycolytic and fatty acid carbon sources for *M. tuberculosis* during hypoxia [39]. In addition, increased succinate production through the TCA cycle and glycoxylate shunt helps to maintain membrane potential, and serves as a substrate for succinate dehydrogenase enzyme to synthesize ATP through oxidative phosphorylation in low oxygen conditions [39]. While the outcome of *M. tuberculosis*, *Saccharomyces cerevisiae*, and *Candida albicans* infections in macrophages significantly differ among these microbes, their initial interaction is relatively similar and they all induce glycxylate cycle very early to fulfill cellular carbon requirement [40]. Our findings suggest that MAB increases glyoxylate and dicarboxylate metabolism in both anaerobic and biofilm conditions to maintain their carbon requirements as well.

Enzymes related to nitrogen metabolic pathways were highly upregulated in MAB of anaerobic and biofilm conditions. It is known that *M. tuberculosis* uses nitrites to reduce to ammonium, which contributes to nitrogen assimilation and, subsequently, promote the pathogen virulence and tolerance under hypoxic stress [41]. Likewise, in the absence of oxygen, *Pseudomonas aeruginosa* uses nitrate or nitrite for respiration [42]. These observations support our finding that nitrogen metabolism might be a crucial component for MAB to survive in anaerobic conditions. Further understanding of nitrogen utilization by MAB could provide new insights into alternative pathways that can be targeted to effectively attenuate the pathogen within the infected host.

Bacteria residing in the extracellular matrix of biofilm face both aerobic and anaerobic conditions. In transcriptional and proteomic studies on diverse environments of *Neisseria gonorrhoeae* biofilm, researchers have described highly expressed pathways of oxidative phosphorylation, pyruvate metabolism, glycolysis/gluconeogenesis, and citrate cycle (TCA cycle) [43] that are also expressed in MAB. In our study, we identified the elevated synthesis of 19 and 10 enzymes of the pyruvate metabolism pathway respectively expressed in MAB of anaerobic and biofilm phenotypes. It has been verified that in an anaerobic environment, *Actinobacillus pleuropneumoniae,* also a respiratory pathogen, increases synthesis of proteins involved in glycolysis and pyruvate metabolisms [44]. In order to maintain extracellular matrix structures, *Staphylococcus aureus* synthesized pyruvate metabolism-related enzymes mainly in the deeper layers of biofilm where the oxygen level is considerably low [45]. The opportunistic pathogen *Pseudomonas aeruginosa* survives under anaerobic conditions by considerably increasing the pyruvate metabolizing enzyme activities [46]. Based on the enriched proteome obtained in this study, we can hypothesize that MAB most likely uses the lactate dehydrogenase, pyruvate formate lyase and/or 2-acetolactase synthase to condensate two molecules of pyruvate to 1-acetoacetate, leading down the pathway line to 2–3 butanediol. In addition, the pyruvate-formate lyase can catalyze pyruvate to acetate.

In this study, we also aimed to uncover common responses expressed by two NTM species, MAB (this study) and MAH [19]. Due to the fact that both pathogens had a limited number of proteins synthesized during exposure to antibiotics under anaerobic and biofilm environments and restricted the metabolic pathway analysis, we focused on exploring the global changes that MAB and MAH undertook in conditions only without drug treatment. We found 19 and 9 metabolic pathways were prominent for both bacterial species in anaerobic and biofilm conditions, respectively. Under anaerobic conditions, both MAH and MAB had highly synthesized enzymes that belong to the oxidative phosphorylation, citrate cycle (TCA cycle), glycolysis/gluconeogenesis, glyoxylate, and dicarboxylate metabolism, nitrogen metabolism, peptidoglycan biosynthesis, glycerophospholipid metabolism, glycerolipid metabolism, and the central metabolism-related pathways.

In order to survive in a low nutrient environment of the biofilm condition, we discovered that both pathogens exploited pathways of the nitrogen metabolism, peptidoglycan biosynthesis, glyoxylate and dicarboxylate metabolism, histidine metabolism, fatty acid metabolism and biosynthesis, and pantothenate and CoA biosynthesis. The activation of the peptidoglycan and fatty acid biosynthesis indicates MAB and MAH attempt to modify the cell surface, influencing bacterial susceptibility to antibiotics, as previously seen in *Mycobacterium avium* subsp. *paratuberculosis* and *Enterococcus faecalis* [47,48].

We expressed the alternative phenotypes in the nonpathogenic *M. smegmatis* of the peptidoglycan biosynthesis, pantothenate and CoA biosynthesis, and nitrogen metabolism pathways displaying importance for MAB and MAH survival under stress conditions and during exposure to antimicrobials. The in vitro and macrophage infection assays demonstrate that the nonpathogenic mycobacteria are able to tolerate the presence of bactericidal concentrations of antimicrobials and prolong bacterial survival for two to four days within phagocytic cells during antibiotic treatments. This observation suggests that the inhibition of the abovementioned pathways most likely will affect MAB and MAH fitness within the host and will prevent development of tolerance phenotypes.

This study establishes global changes in the synthesis of many enzymes of MAB, promoting metabolic shifts and enhancing bacterial persistence within different environments. It also demonstrates how the pathogen protects itself against the action of antimicrobials. It highlights that mechanisms used in many instances are common between two NTM species, and the majority of proteins identified in this study connect several different pathways, exposing some survival strategies employed by both respiratory pathogens.

## Figures and Tables

**Figure 1 microorganisms-08-00698-f001:**
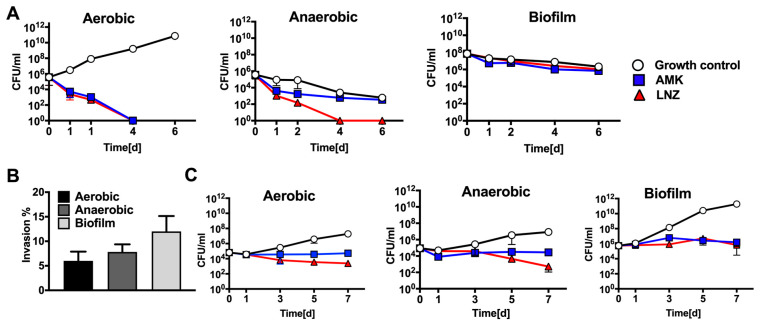
Antibiotic killing kinetics in vitro and in cultured macrophages. (**A**) Time-kill curves of *M. abscessus* subsp. *abscessus* (MAB) under aerobic, anaerobic and biofilm conditions and using bactericidal concentrations of amikacin (AMK) and linezolid (LNZ) show viable bacterial colony forming units (CFUs) over 6 days. Antibiotics were added to bacterial culture in 7H9 broth at time zero, and drug treatment groups were compared to MAB growth control. (**B**) Macrophage invasion assay demonstrates that MAB of biofilm phenotype invaded cells at a higher rate than bacteria of aerobic and anaerobic phenotypes. THP-1 cells were infected as described in the materials and methods, and the percentage of invasion was calculated as the percentage of intracellular bacteria recovered at 2 h infection to initially applied inoculum. (**C**) MAB survival rates in THP-1 macrophages display bacterial number of the aerobic, anaerobic, or biofilm phenotype recorded over 7 days with and without AMK and LNZ treatments. Antimicrobials were added to culture monolayers at 2 h of infection and then every alternate day. Growth control without drug treatment is also shown.

**Figure 2 microorganisms-08-00698-f002:**
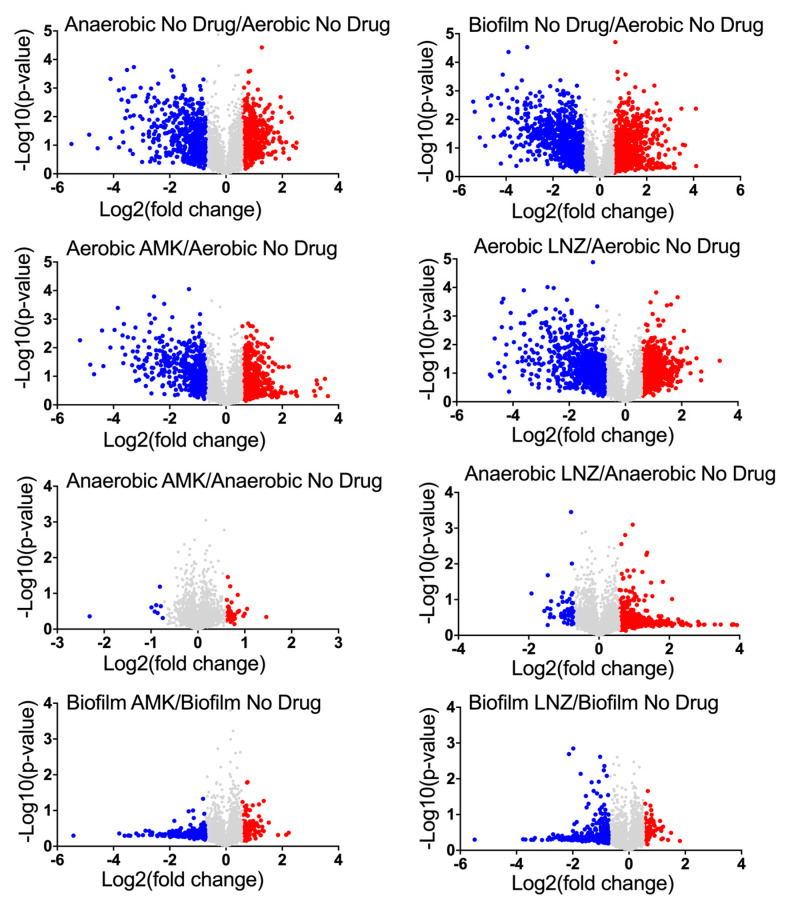
Volcano plots displaying MAB induced or repressed proteins with and without drug treatments. This plot displays the log2 changes in protein synthesis induced under anaerobic and biofilm conditions when compared with the aerobic control. Protein enrichments of AMK or LNZ treated MAB are compered to proteome obtained under the corresponding condition. Synthesized proteins ≥1.5-fold higher than control are shown in red, and ≥1.5-fold lower than control are in blue; each dot is one protein.

**Figure 3 microorganisms-08-00698-f003:**
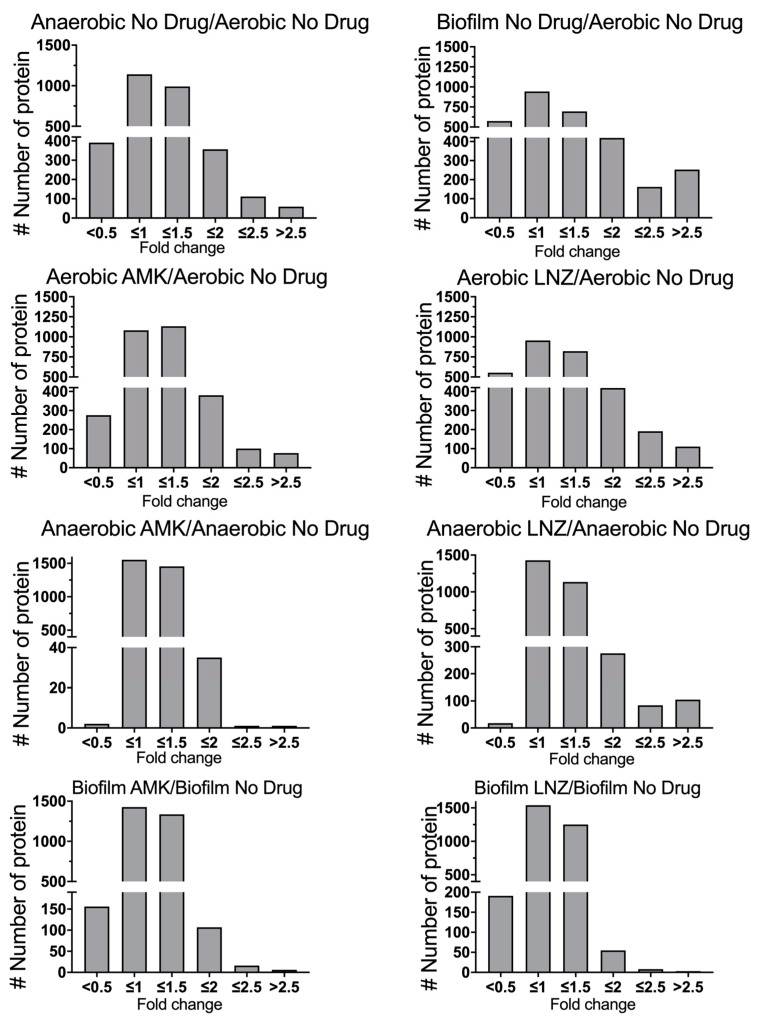
Fold changes of differentially expressed MAB proteins. The histograms demonstrate the distributions of fold changes of differentially expressed proteins in anaerobic and biofilm conditions with and without AMK and LNZ treatments.

**Figure 4 microorganisms-08-00698-f004:**
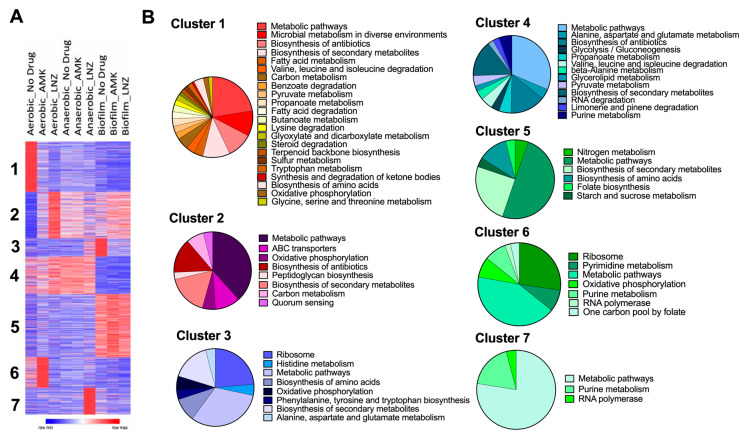
The heatmap and clustering analysis. (**A**) The heatmap demonstrates clustering of different protein groups under stress conditions with various enrichment levels in antibiotic or no antibiotic treatment groups. The color reflects the degree of change. K-means clustering was performed for 3722 genes across all studied groups (adjusted *p*-value < 0.05), and K = 7 was determined as an optimal number. (**B**) Pie charts correspond to 7 clusters of the heat map and display enriched metabolic pathways of the KEGG (Kyoto Encyclopedia of Genes and Genomes) database.

**Figure 5 microorganisms-08-00698-f005:**
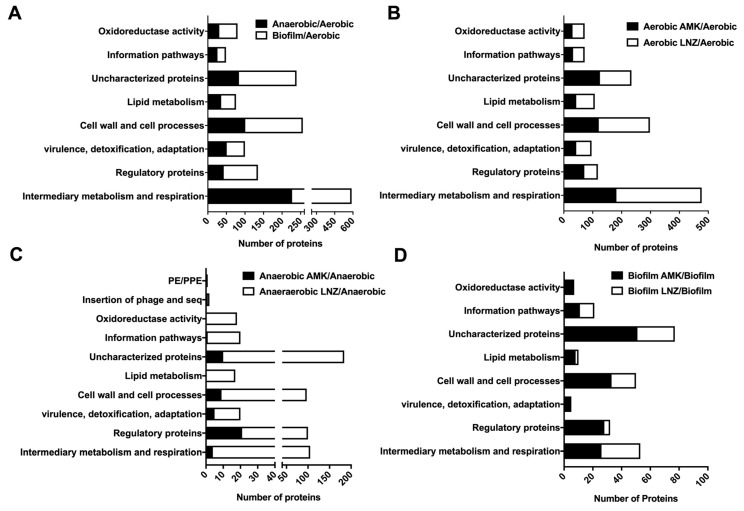
Functional classification of MAB upregulated proteins by ≥1.5-fold. Enriched proteins of (**A**) anaerobic and biofilm conditions; (**B**) aerobic condition during exposure to AMK or LNZ; (**C**) anaerobic condition during exposure to AMK or LNZ; and (**D**) biofilm condition during exposure to AMK or LNZ. The analysis was done by finding MAB protein homologs in *M. tuberculosis* H37Rv strain and using the functional categorization available on TubercuList webserver of Institute Pasteur. MAB proteins that did not show any homology were classified based on predicted or known function for conserved domain/motif search at NCBI (National Center for Biotechnology Information) and the Pfam database through the Sanger Institute. Histograms show number of proteins belonging to each functional category.

**Figure 6 microorganisms-08-00698-f006:**
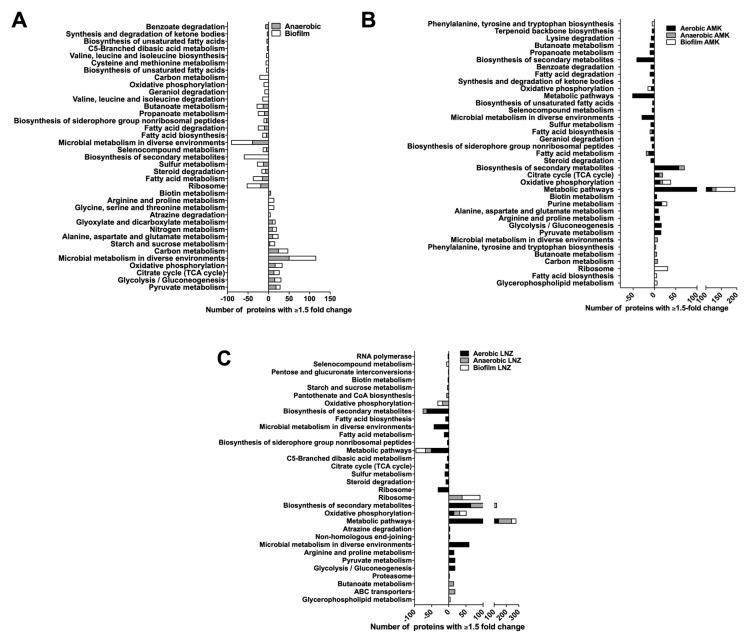
Metabolic pathway enrichment. Proteins were grouped based on the KEGG pathway database for enriched and repressed proteins in (**A**) anaerobic and biofilm, (**B**) AMK treated aerobic, anaerobic and biofilm, and (**C**) LNZ treated aerobic, anaerobic and biofilm experimental groups.

**Figure 7 microorganisms-08-00698-f007:**
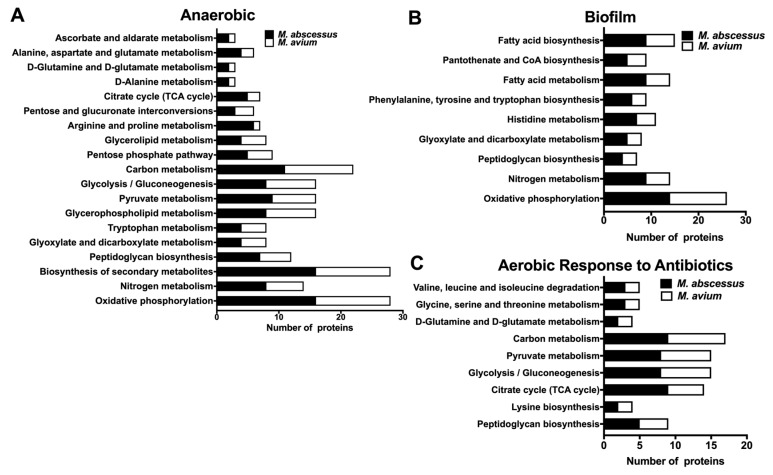
MAB and *M. avium* subsp. *hominissuis* (MAH) shared metabolic pathway enrichment. MAB and MAH proteins synthesized ≥1.5-fold over control were matched by aligning the amino acid sequences, and then grouped into metabolic pathways using the KEGG database for upregulated proteins in (**A**) anaerobic, (**B**) biofilm and (**C**) aerobic conditions for MAB treated with AMK and LNZ, and MAH treated with AMK and clarithromycin (CLA).

**Figure 8 microorganisms-08-00698-f008:**
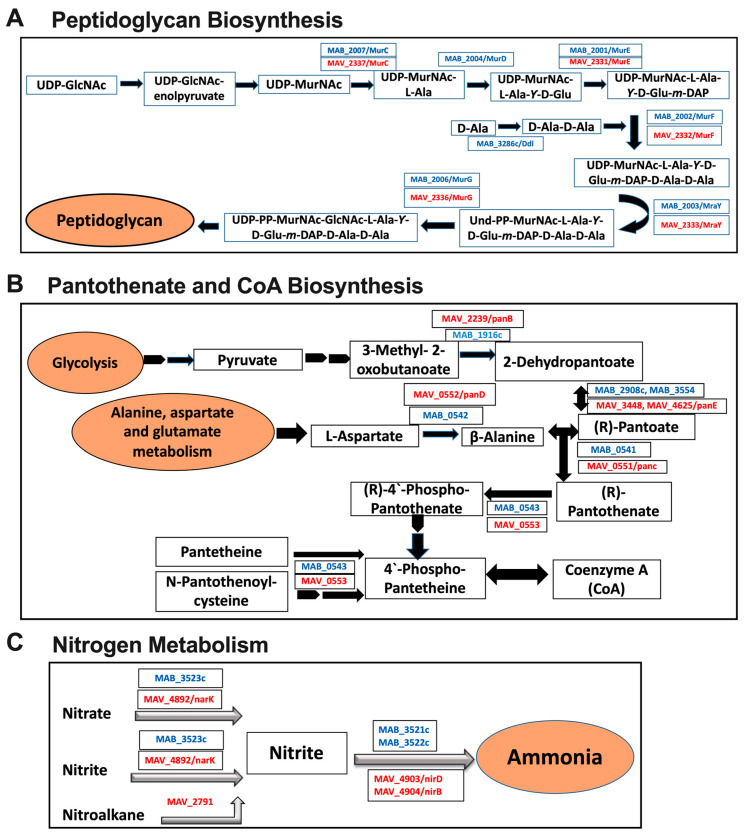
MAH and MAB common enzymes associated with metabolic pathways. (**A**) Peptidoglycan biosynthesis pathway upregulated under anaerobic and biofilm conditions, and during exposure with antibiotics. (**B**) Pantothenate and CoA biosynthesis and (**C**) Nitrogen metabolism pathways upregulated under anaerobic and biofilm conditions. MAB and MAH homologous proteins are listed. MAB proteins are marked in blue and MAH in red.

**Figure 9 microorganisms-08-00698-f009:**
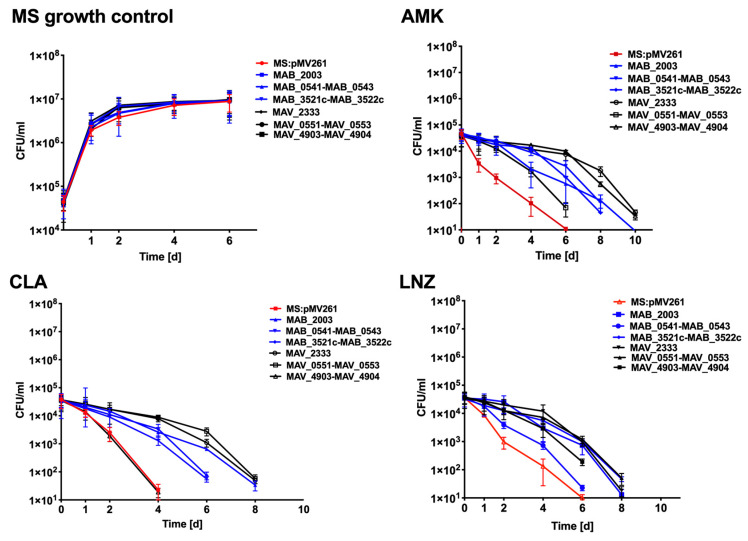
In vitro time-kill curves for *M. smegmatis* clones expressing MAB and MAH homologues genes of common metabolic pathways. The *MAB_2003* and *MAV_2333* genes of peptidoglycan biosynthesis pathway; *MAB_0541*, *MAB_0542*, *MAB_0543* and *MAH_0551*, *MAV_0552*, *MAV_0553* genes of pantothenate and CoA biosynthesis pathway; *MAB_3521c*, *MAB_3522c* and *MAV_4903*, *MAV_4904* genes of the nitrogen metabolism pathway.

**Figure 10 microorganisms-08-00698-f010:**
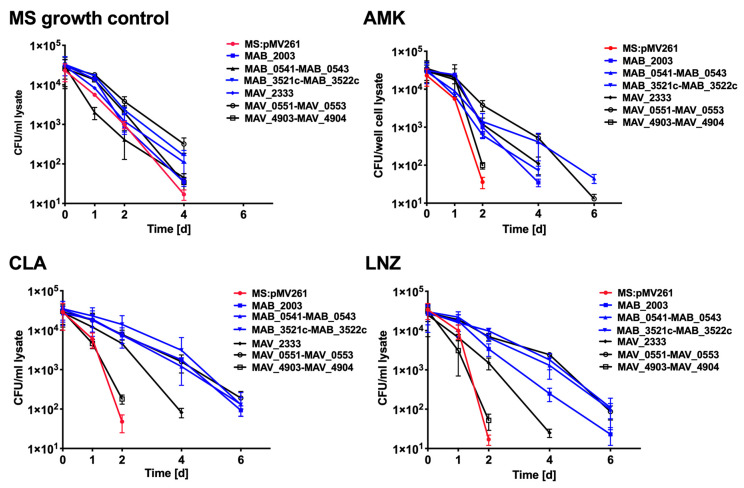
The killing dynamics of *M. smegmatis* expressing MAB and MAH homologous genes of common metabolic pathways in THP-1 macrophages.

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
