# Peer review of "Exposure of Mycobacterium abscessus to Environmental Stress and Clinically Used Antibiotics Reveals Common Proteome Response among Pathogenic Mycobacteria"

_microorganisms, 2020, doi:10.3390/microorganisms8050698_

Round 1
Reviewer 1 Report
The manuscript entitled “Exposure of Mycobacterium abscessus to environmental stress and clinically used antibiotics reveals common proteome response among pathogenic mycobacteria” is focused on the proteomic analysis of Mycobacterium abscessus subsp. abscessum (MAB) in biologically relevant conditions and the comparison with M. avium subsp. hominissuis (MAH). The authors reported MAB proteomics under aerobic, anaerobic and biofilm conditions either in presence or in absence of the antimicrobials linezolid (LNZ) and amikacin (AMK). The identification of the metabolic pathways activated or deactivated under these biologically relevant conditions could provide relevant information to fight MAB bacterial infections. The comparison with former proteomic analysis performed on MAH under similar conditions highlights the presence of common activated metabolic pathways. MAB is a clinically relevant mycobacterium resistant to antibiotic treatments and responsible for pulmonary infections. Stating the importance to develop new effective treatments to fight MAB infections, the present manuscript provides relevant insights exploitable for the identification of new bacterial targets.
I will recommend the acceptance after minor revisions (listed below).
Minor revisions:
- Introduction, lines 44-46. The spectrum of antibiotic resistance should be specified.
- Introduction, lines 47-55. The activities of the antimicrobials AMK and LNZ, used as reference drugs in the present study, should be briefly introduced (description of their mechanisms of action and main bacterial targets).
- Materials and Methods, line 79. The strain used in the present study is MAB 19977. The terms MAB and MAB 19977 are used throughout the manuscript for this strain. The authors should use a unique name to avoid confusion.
- Materials and Methods, line 103. Change h in days, consistently with the sections describing the results.
- Materials and Methods, line 128. Correct “Next, Infected” in “Next, infected”
- Materials and Methods, line 151. The method by which the total protein concentrations were determined should be specified together with the device.
- Materials and Methods, line 202. Change “analysis was” in “analyses were”
- Materials and Methods, line 252. Figure 8 is the first figure cited in the manuscript. The figure numbering should be consistent with their progressive citation throughout the manuscript.
- Materials and Methods, line 262. Please specify the unit after “106”
- Materials and Methods, line 264. Please replace 24 h and 48 h with 1 and 2 (days).
- Results, section 3.1. The authors should explain why they have decided to use the antimicrobials AMK and LNZ in the studies.
- Results, lines 287-291. BC values for both AMK and LNZ are > 10 times higher than MIC values, the authors should provide an explanation for these results.
- Figure 1. The time in x axis should be expressed in days consistently with the manuscript. Furthermore, to facilitate the comparison among the different conditions, the same scale should be used in the y axis of all panels.
- Results, lines 323-326. The sentence is not fully clear, please rewrite it in a clearer form.
- Figure 2. Please move it before Figure 3
- Figure 3. To facilitate the comparison among the different conditions, the same scale should be used in the y axis of all panels.
- Results, line 407. Change 5C in 5D.
- Results, lines 426-429. The sentence is too long and difficult to read. The authors should divide it in two sentences.
- Results, line 437. Correct “were increases” in “were increased”
- Results, section 3.6. The authors should explain why they are comparing these two bacterial strains.
- Results, lines 456-458. In Figure 7A, 19 common pathways are displayed instead of the 20 mentioned in the sentence.
- Results, lines 464-475. Since both MAH and MAB were treated with a common antimicrobial AMK, it would be interesting to evaluate the effect of this antimicrobial alone on both strains. The author should explain why they are comparing the results obtained using different combinations of antimicrobial agents instead of using the same.
- Results, lines 468-469. In Figure 7C, 9 common pathways are displayed instead of the 11 mentioned in the sentence.
- Figure 9 and 10. The time in x axis should be expressed in days consistently with the manuscript. Furthermore, to facilitate the comparison among the different conditions, the same scale should be used in the y axis of all panels.
- Discussion, line 548. Name the metabolic changes already implied to prolong bacterial survival.
- Discussion, line 572. Change “Transcriptional” with “In transcriptional”
- Discussion, line 583-586. These sentences are not supported by the results in which only the enzyme/proteins levels are analyzed. The suggested metabolic pathways deserve supporting data otherwise they are just speculations.
Author Response
We appreciate the reviewer’s thorough review of our paper and suggestions. Thank you.
Minor revisions:
- Introduction, lines 44-46. The spectrum of antibiotic resistance should be specified.
A: We modified a sentence, highlighting MAB intrinsic and acquired resistance ability, and there is no need for antibiotic resistance spectrum clarification in this paragraph.
- Introduction, lines 47-55. The activities of the antimicrobials AMK and LNZ, used as reference drugs in the present study, should be briefly introduced (description of their mechanisms of action and main bacterial targets)
A: We added brief description of antibiotic actions with the citation.
- Materials and Methods, line 79. The strain used in the present study is MAB 19977. The terms MAB and MAB 19977 are used throughout the manuscript for this strain. The authors should use a unique name to avoid confusion.
A: As suggested by the reviewer, we decided to use MAB name throughout the text.
- Materials and Methods, line 103. Change h in days, consistently with the sections describing the results.
A: We changed hours to days.
- Materials and Methods, line 128. Correct “Next, Infected” in “Next, infected”
A: The paragraph has been modified.
- Materials and Methods, line 151. The method by which the total protein concentrations were determined should be specified together with the device.
A: The method is mentioned at the end of a paragraph.
- Materials and Methods, line 202. Change “analysis was” in “analyses were”
A: We made a change.
- Materials and Methods, line 252. Figure 8 is the first figure cited in the manuscript. The figure numbering should be consistent with their progressive citation throughout the manuscript.
A: We removed Figure 8 citation and changed the text accordingly.
- Materials and Methods, line 262. Please specify the unit after “106”
A: We added CFU/ml, thank you.
- Materials and Methods, line 264. Please replace 24 h and 48 h with 1 and 2 (days).
A: We made this change.
- Results, section 3.1. The authors should explain why they have decided to use the antimicrobials AMK and LNZ in the studies.
A: We added a sentence clarifying why we used AMK and LNZ in this study.
- Results, lines 287-291. BC values for both AMK and LNZ are > 10 times higher than MIC values, the authors should provide an explanation for these results.
A: MIC and BC results obtained in our study are aligned with already published data and it is no surprise that M. abscessus BC concentrations at which 99.9% bacteria are cleared are significantly higher than MIC. We added a sentence just to highlight no significant growth up to 16X MIC for both antibiotics.
- Figure 1. The time in x axis should be expressed in days consistently with the manuscript. Furthermore, to facilitate the comparison among the different conditions, the same scale should be used in the y axis of all panels.
A: We made corrections as suggested by the reviewer.
- Results, lines 323-326. The sentence is not fully clear, please rewrite it in a clearer form.
A: We modified sentence.
- Figure 2. Please move it before Figure 3
A: We placed figure 2 before Figure 3.
- Figure 3. To facilitate the comparison among the different conditions, the same scale should be used in the y axis of all panels.
A: Y-axis panels are now adjusted.
- Results, line 407. Change 5C in 5D.
A: We made this change. Thank you.
- Results, lines 426-429. The sentence is too long and difficult to read. The authors should divide it in two sentences.
A: We simplified the sentence.
- Results, line 437. Correct “were increases” in “were increased”
A: We corrected.
- Results, section 3.6. The authors should explain why they are comparing these two bacterial strains.
A: We added explanation.
- Results, lines 456-458. In Figure 7A, 19 common pathways are displayed instead of the 20 mentioned in the sentence.
A: We corrected.
- Results, lines 464-475. Since both MAH and MAB were treated with a common antimicrobial AMK, it would be interesting to evaluate the effect of this antimicrobial alone on both strains. The author should explain why they are comparing the results obtained using different combinations of antimicrobial agents instead of using the same.
A: The goal of this study was to identify common pathways and targets in both NTM pathogens promoting bacterial tolerance mechanism under the pressure of antibiotics (regardless of the drug). We clarified this in the text.
- Results, lines 468-469. In Figure 7C, 9 common pathways are displayed instead of the 11 mentioned in the sentence.
A: We made correction. Thank you.
- Figure 9 and 10. The time in x axis should be expressed in days consistently with the manuscript. Furthermore, to facilitate the comparison among the different conditions, the same scale should be used in the y axis of all panels.
A: As suggested by the reviewer, we made corrections and adjusted the scale within in vitro and in cultured macrophages separately.
- Discussion, line 548. Name the metabolic changes already implied to prolong bacterial survival.
A: We added citation for the review paper.
- Discussion, line 572. Change “Transcriptional” with “In transcriptional”
A: We changed it.
- Discussion, line 583-586. These sentences are not supported by the results in which only the enzyme/proteins levels are analyzed. The suggested metabolic pathways deserve supporting data otherwise they are just speculations.
A: The reviewer is right. We changed this sentence to: ”Based on the enriched proteome obtained in this study, we can hypothesize that…”
Reviewer 2 Report
In this paper, the authors found out that under anaerobic and biofilm conditions, non-tuberculous mycobacterium (NTM)‑ Mycobacterium abscessus subsp. abscessus (MAB) when treated with fist line antibacterial drugs Amikacin and linezolid overexpress different enzymes necessary for its survival under these conditions. These proteomic results will help to design new drugs targeting previously unknown enzymes which are overexpressed under hypoxic and biofilm conditions to inhibit the bacterial cell growth.
Minor comments:
1. The authors should discuss targets of amikacin, linezolid, and clarithromycin in the introduction section. It will help to understand their mechanism of action.
2. Authors could cite the following paper in an introduction section
Nat Rev Microbiol. 2020 Feb 21. doi: 10.1038/s41579-020-0331-1.
Non-tuberculous mycobacteria and the rise of Mycobacterium abscessus.
3. Page 9, switch figures 2 and 3.
4. How many over synthesized proteins were similar between AMK and LNZ treated MAB cells?
5. Did they treat MAB with different combinations of AMK/LNZ/CLA to study proteome?
6. Page 1, line 42-43, size of font is different
Author Response
Reviewer 2
- The authors should discuss targets of amikacin, linezolid, and clarithromycin in the introduction section. It will help to understand their mechanism of action.
A: We added a sentence and citation (6).
- Authors could cite the following paper in an introduction section. Nat Rev Microbiol. 2020 Feb 21. doi: 10.1038/s41579-020-0331-1. Non-tuberculous mycobacteria and the rise of Mycobacterium abscessus.
A: We added this reference, Thank you.
- Page 9, switch figures 2 and 3.
A: We made this correction.
- How many over synthesized proteins were similar between AMK and LNZ treated MAB cells?
A: Similar proteins found between AMK and LNZ are 73 in the aerobic, 1 in biofilm and none in anaerobic condition.
- Did they treat MAB with different combinations of AMK/LNZ/CLA to study proteome?
A: No, we did not treated MAB with combinations of drugs. The bacterial proteome is presented for each drug separately under aerobic, anaerobic or biofilm conditions.
- Page 1, line 42-43, size of font is different
A: We changed the font size. Thank you.
Reviewer 3 Report
The manuscript of Rojony R. and collaborators reports the investigation of antibiotic susceptibility of the emergent human M. abscessusunder different growth conditions. The authors performed also very interesting proteomic data obtained on M. abscessusunder aerobic, anaerobic and biofilm conditions. This work reveals interesting changes in several metabolic pathways. The transfer of the identified upregulated protein in stress conditions from M. abscessusto M. smegmatisconfer to the latter antibiotic resistance. The proteomic data are further compared to those obtained previously from M. aviumunder similar which reveals overproduction of several protein orthologues.
Overall this study concerning M. abscessus known as a highly drug resistant bacteria and emergent human pathogen is very interesting, it brings new knowledge into the adaptation of M. abscessusto environmental changes. The manuscript is also pretty well written.
However, modifications in some aspects of this study have to be performed prior to publication.
1) A recent excellent review on M. abscessus could be added in the introduction: Johansen et al., Nat Rev Microbiol. 2020 Feb 21. doi: 10.1038/s41579-020-0331-1
2) Lane 128 change amikacin to AMK
3) Figure 1 : add color tothe curves as it is difficult to distinguish them, the signs are very small
4) Like many other authors in the field of M. abscessus, the author of this study did not distinguish between the smooth and rough morphotype. However, the two morphotypes behave differently particularly when it comes to the capacity to form biofilms. The smooth morphotype seems to be more prone to form biofilm compare with the rough one. Do the author know what type of colonies they have in their biofilm. This is important as the S morphotype is supposed to be the colonizing form and the R the virulent one. It could be interesting to discuss this.
5) As M. abscessus infections are very problematic to treat, it is a bit of a pity that the authors are not discussing much more about the enzymes that know to triggers resistance. For example WhiB7 is very well known to be upregulated when M. abscessus is challenged with antibiotics, particularly aminoglycosides. What about efflux pump? Do the authors see some these proteins more expressed their proteomic data, if yes please name them ? Discussing these aspects would add value to this MS and interest a broader readership.
6) For clarity, could the authors emphasize a bit more the role/function of the protein that were transferred to M. smegmatis.
7) Comment: the use of AMK might not be that appropriate to test intracellular killing as it is know that aminoglycosides are actually poorly crossing biological membranes.
Author Response
Reviewer 3
- A recent excellent review on M. abscessus could be added in the introduction: Johansen et al., Nat Rev Microbiol. 2020 Feb 21. doi: 10.1038/s41579-020-0331-1
A: We added the suggested reference.
- Lane 128 change amikacin to AMK
A: We made a change.
- Figure 1 : add color to the curves as it is difficult to distinguish them, the signs are very small
A: We added color and increased the size of symbols as suggested by the reviewer.
- Like many other authors in the field of abscessus, the author of this study did not distinguish between the smooth and rough morphotype. However, the two morphotypes behave differently particularly when it comes to the capacity to form biofilms. The smooth morphotype seems to be more prone to form biofilm compare with the rough one. Do the author know what type of colonies they have in their biofilm. This is important as the S morphotype is supposed to be the colonizing form and the R the virulent one. It could be interesting to discuss this.
A: The clinical isolate MAB 19977 has a smooth phenotype. This strain is virulent and can colonize, invade and survive in cultured cells.
- As abscessus infections are very problematic to treat, it is a bit of a pity that the authors are not discussing much more about the enzymes that know to triggers resistance. For example WhiB7 is very well known to be upregulated when M. abscessus is challenged with antibiotics, particularly aminoglycosides. What about efflux pump? Do the authors see some these proteins more expressed their proteomic data, if yes please name them ? Discussing these aspects would add value to this MS and interest a broader readership.
A: The central goal of this project was to identify alternative pathways that allow the pathogen to survive under different environmental stresses and during treatment with bactericidal concentrations of anti-NTM drugs. The mechanism of drug action is well studied for MAB as the reviewer points out. In this study, we found that MAB undergo significant metabolic changes, and by profiling the proteome response, we aimed to identify targets that can possibly help fast killing of the pathogen. We succeeded it by overexpressing these genes in M. smegmatis, which prolonged bacterial survival during drug treatments in vitro as well as in cultured macrophages.
- For clarity, could the authors emphasize a bit more the role/function of the protein that were transferred to smegmatis.
A: We added information on gene functions in the results section.
- Comment: the use of AMK might not be that appropriate to test intracellular killing as it is know that aminoglycosides are actually poorly crossing biological membranes.
A: We agree with a reviewer, however, this antibiotic is widely used to treat MAB patients in clinics.